# A decade of theory as reflected in *Psychological Science* (2009–2019)

**Jonathon McPhetres** [1]*, **Nihan Albayrak-Aydemir** [2], **Ana Barbosa Mendes** [3], **Elvina C. Chow** [4‡], **Patricio Gonzalez-Marquez** [5], **Erin Loukras** [5‡], **Annika Maus** [6], **Aoife O'Mahony** [7], **Christina Pomareda** [8‡], **Maximilian A. Primbs** [9‡], **Shalaine L. Sackman** [10‡], **Conor J. R. Smithson** [11], **Kirill Volodko** [5]

1 Durham University, Durham, United Kingdom, 2 London School of Economics and Political Science, London, United Kingdom, 3 ITEC, Faculty of Psychology and Educational Sciences, KU Leuven, Leuven, Belgium, 4 Pepperdine University, Malibu, California, United States of America, 5 Quest University, Squamish, Canada, 6 University of Cambridge, Cambridge, United Kingdom, 7 Cardiff University, Cardiff, United Kingdom, 8 University of Birmingham, Birmingham, United Kingdom, 9 Radboud University, Nijmegen, Netherlands, 10 University of Regina, Regina, Canada, 11 Vanderbilt University, Nashville, Tennessee, United States of America

☯ These authors contributed equally to this work.
‡ These authors also contributed equally to this work.
* jon.mcphetres@gmail.com

**Data Availability Statement:** All relevant data are available from the Open Science Framework (OSF) database (osf.io/hgn3a). The OSF preregistration is also available (osf.io/d6bcq/).

## Abstract

The dominant belief is that science progresses by testing theories and moving towards theoretical consensus. While it's implicitly assumed that psychology operates in this manner, critical discussions claim that the field suffers from a lack of cumulative theory. To examine this paradox, we analysed research published in *Psychological Science* from 2009–2019 ($N = 2,225$). We found mention of 359 theories in-text, most were referred to only once. Only 53.66% of all manuscripts included the word *theory*, and only 15.33% explicitly claimed to test predictions derived from theories. We interpret this to suggest that the majority of research published in this flagship journal is not driven by theory, nor can it be contributing to cumulative theory building. These data provide insight into the kinds of research psychologists are conducting and raises questions about the role of theory in the psychological sciences.

> *"The problem is almost anything passes for theory."* -Gigerenzer, 1998, pg. 196 (1).

## Introduction

Many have noted that psychology lacks the cumulative theory that characterizes other scientific fields [1–4]. So pressing has this deficit become in recent years that many scholars have called for a greater focus on theory development in the psychological sciences [5–11].

At the same time, it has been argued that there are perhaps *too many theories* to choose from [3, 12–14]. One factor contributing to this dilemma is that theories are often vague and poorly specified [2, 15], so a given theory is unable to adequately explain a range of phenomena without relying on rhetoric. Thus, psychology uses experimentation to tell a narrative rather

**Funding:** The author(s) received no specific funding for this work.

**Competing interests:** The authors have declared that no competing interests exist.

than to test theoretical predictions [16, 17]. From this perspective, psychology needs more exploratory and descriptive research before moving on to theory building and testing [18–20].

Despite these competing viewpoints, it is often claimed that psychological science follows a hypothetico-deductive model like most other scientific disciplines [21]. In this tradition, experiments exist to test predictions derived from theories. Specifically, researchers should be conducting *strong tests* of theories [22–24] because strong tests of theory are the reason some fields move forward faster than others [2, 4, 25]. That is, the goal scientists should be working towards is theoretical consensus [1, 2, 26–28]. At a glance, it would appear that most psychological research proceeds in this fashion, because papers often use theoretical terms in introduction sections, or name theories in the discussion section. However, no research has been undertaken to examine this assumption and what role theory actually plays in psychological research.

So, which is it? If there is *a lack of theory*, then most articles should be testing a-theoretical predictions or conducting descriptive and exploratory research. If there is *too much theory*, then almost every published manuscript should exist to test theoretically derived predictions.

To examine the role of theory in psychological research, we analysed articles published from 2009–2019 in the journal *Psychological Science*. We use this data to answer some specific questions. First, we are interested in distinguishing between specific and casual uses of theory. So, we analyse how often theory-related words are used overall and how often a specific theory is named and/or tested. Additionally, given that preregistration can help prevent HARKING [29], we examine whether articles that name and/or test a theory are more likely to be preregistered. Next, it's possible that some subsets of psychological research might be more or less reliant on theory. To examine this, we investigate whether studies that name and/or test a theory are more likely to generate a specific kind of data. Finally, to provide greater context for these analyses, we examined how many theories were mentioned over this time period and how many times each was mentioned.

## Disclosures

All analyses conducted are reported and deviations are disclosed at the end of this section. Our sample size was pre-determined and was based on the entire corpus of published articles. Finally, because this research does not involve human subjects, ethics approval was not sought.

## Materials and methods

We accessed all the articles published in *Psychological Science* from 2009–2019. We chose this journal because it is the flagship journal of the Association for Psychological Science and one of the top journals in the field that publishes a broad range of research from all areas of the discipline. Additionally, this journal explicitly states that theoretical significance is a requirement for publication [30, 31].

As preregistered https://osf.io/d6bcq/?view_only=af0461976df7454fbcf7ac7ff1500764, we excluded comments, letters, errata, editorials, or other articles which did not test original data because they could not be coded or because, in some cases, they were simply replications or re-analyses of previously published articles. This resulted in 2,225 articles being included in the present analysis.

## Definition

Many useful definitions and operationalisations of a *scientific theory* have been put forward [4, 32–34] and we drew on these for the present work. The definition of a *scientific theory* for the purposes of this research is as follows:

A theory is a framework for understanding some aspect of the natural world. A theory often has a name—usually this includes the word *theory*, but may sometimes use another label (e.g., model, hypothesis). A theory can be specific or broad, but it should be able to make predictions or generally guide the interpretation of phenomena, and it must be distinguished from a *single effect*. Finally, a theory is not an untested prediction, a standard hypothesis, or a conjecture.

We used this definition in order to distinguish its use from colloquial and general uses of the word, not to evaluate the strength, viability, or suitability of a theory.

## Text mining

Article PDFs were first mined for the frequency of the words *theory*, *theories*, and *theoretical* using the TM [35] and Quanteda [36] packages in R. Word frequencies were summed and percentages were calculated for each year and for the entire corpus. We did not search or code for the terms *model* or *hypothesis* because these are necessarily more general and have multiple different meanings, none of which overlap with *theory* (but see the Additional Considerations section for more on this).

## Coding

After identifying the articles that used the words *theory* and *theories*, 10 trained coders further examined those articles. Instances of the word *theoretical* were not examined further because it is necessarily used generally (and because it was used less than, but often alongside, *theory* and *theories*).

Each article was initially scored independently by two individual coders who were blind to the purpose of the study; Fleiss' Kappa is report for this initial coding. Recommendations suggest that a kappa between .21-.40 indicates fair agreement, .41-.60 indicates moderate agreement, .61-.80 indicates substantial agreement, and .81–1.0 is almost perfect agreement [37].

After the initial round of coding, two additional blind coders and the first author each independently reviewed a unique subset of disagreements to resolve ties. This means that the ratings we analyse in the following section are the result of ratings only for which two independent coders (or two out of three coders) agreed 100%.

For each article, the following categories were coded:

**Was a specific theory referred to by name?.** For each article, the coder conducted a word-search for the string "theor" and examined the context of each instance of the string. We recorded whether each paper, at any point, referred to a specific theory or model by name. Instances of words in the reference section were not counted nor coded further. General references to theory (e.g., psychological theory) or to classes or groups of theories (e.g. relationship theories) were not counted because these do not allow for specific interpretations or predictions. Similarly, instances where a theory, a class of theories, or an effect common across multiple studies was cited in-text along with multiple references but not named explicitly—for example, "cognitive theory (e.g. Author A, 1979; Author B, 1996; Author C & Author D, 2004) predicts"—were also not counted because these examples refer to the author's own interpretation of or assumptions about a theory rather than a specific prediction outlined by a set of theoretical constraints. Initial coder agreement was 78% (and significantly greater than chance, Fleiss' kappa = .45, $p < .001$).

**Did the article claim to test a prediction derived from a specific theory?.** For each article, the coder examined the abstract, the section prior to introducing the first study, the results, and the beginning of the general discussion. We recorded whether the paper, at any point,

*explicitly claimed* to test a prediction derived from a specific theory or model. As above, this would have been needed to be made clear by the authors to avoid categorising general predictions, auxiliary assumptions, indirect and verbal interpretations of multiple theories, models, or hypotheses derived from personal expectations as being theoretically derived. Initial coder agreement was 74% (and significantly greater than chance, Fleiss' kappa = .24, $p < .001$).

**What was the primary type of data generated by the study?.** For each article, the coder examined the abstract, the section prior to introducing the first study, the results, and the beginning of the general discussion. The primary type of data used in the study was coded as either self-report/survey, physiological/biological, observational/behavioural (including reaction times), or other. In the case of multiple types of data across multi-study papers, we considered the abstract, the research question, the hypothesis, and the results in order to determine the type of data most relevant to the question. Initial coder agreement was 64% (and significantly greater than chance, Fleiss' kappa = .42, $p < .001$).

**Did the article include a preregistered study?.** Preregistration is useful for restricting HARKing [29]. It is also useful for testing pre-specified and directional predictions, and hypotheses derived from competing theories. As such, we reasoned that preregistered studies may be more likely to test theoretical predictions.

We coded whether the article included a preregistered study. This was identified by looking for a badge as well as conducting a word search for the strings "prereg" and "pre-reg". Initial coder agreement was 99% (and significantly greater than chance, Fleiss' kappa = .97, $p < .001$).

## Theory counting

The number of theories named directly in the text were recorded and summed by year to provide an overview of how frequently each theory was invoked. The goal of this was to simply create a comprehensive list of the names and number of theories that were referred to in the text at any point. To be as inclusive as possible, slightly different classification criteria were used (see S1 File).

## Transparency statement

Our original preregistered analysis plan did not include plans for counting the total number of theories mentioned in text, nor for examining the frequency of the words *model* and *hypothesis*. Additionally, coding the instances of the word *hypothesis* was not preregistered, but was added after a round of reviews. Finally, for simplicity, we have focused on percentages out of the total articles coded (rather than presenting separate percentages for frequencies of *theory* and *theories*); complete counts and percentages are presented in the S1 File.

## Results

### Question 1: How often are theory-related words used?

To begin, the complete corpus of articles was analysed ($N = 2,225$). Between the years 2009 and 2019, the word *theory* was used in 53.66% of articles, the word *theories* was used in 29.80% of articles, and the word *theoretical* was used in 32.76% of articles (note that these categories are non-exclusive). Total percentages and raw counts by year are presented in the S1 and S2 Tables in S1 File.

### Question 2: How often was a theory named and/or tested?

The 1,605 articles including the word *theory* or *theories* were further coded to examine the context of the word. Of these articles, only 33.58% of them named a specific theory—that is,

66.42% used the word superfluously. Further, only 15.33% of the 1,605 articles explicitly claimed to test a prediction derived from a theory.

To put this differently, only 24.22% of all the articles published over the 11-year period (*N* = 2,225) actually named a specific theory in the manuscript. For example, they used "psychological theory" or "many theories. . ." instead of naming and citing a specific theory. This means that the remainder of those papers either 1) did not derive predictions, operationalisations, analytic strategies, and interpretations of their data from theory, or 2) did not credit previous theory for this information.

The words *theories* and *theoretical* showed similar patterns, but they were used less often than the word *theory*; for simplicity, we present a detailed summary of these counts by year in the S2 Table in S1 File. The pattern of these effects by year is depicted in Fig 1, below.

### Question 3: Are articles that name a specific theory more likely to be preregistered?

Because there were no preregistered articles prior to 2014, we considered only articles published from 2014 onwards (N = 737) for this part of the analysis. Articles that named a specific theory were no more or less likely to be preregistered. Specifically, 11.11% of articles that explicitly named a specific theory were preregistered. In contrast, 11.31% of articles that did not name a theory were preregistered.

Conversely, articles that actually tested a specific theory were only slightly more likely to be preregistered. Of the articles that were preregistered, 15.66% stated that they tested a specific

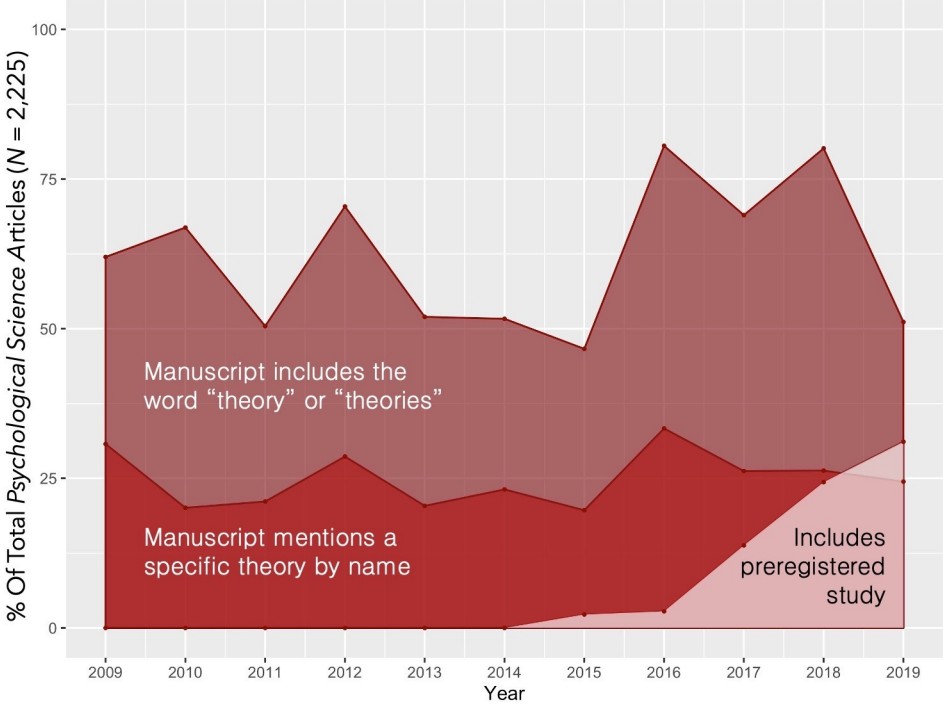

**Fig 1. Percentage of total *Psychological Science* articles from 2009–2019 that use the word theory, name a specific theory, and include a preregistered study.** The percentage of articles that included the words theory/theories, mentioned a theory by name, and were preregistered was calculated out of the total number of articles published from 2009–2019 in *Psychological Science* excluding comments, editorials, and errata (*N* = 2,225); note that for simplicity this figure counts all articles that received a preregistered badge (even if they were not coded in the present study).

theory. Of the articles that were not preregistered, 12.84% stated that they tested a specific theory. See S3 and S4 Tables in S1 File for full counts by year.

## Question 4: Are studies that name and/or test theories more likely to generate a specific kind of data?

Of the 1,605 articles coded over the 11-year period, the overwhelming majority (55.26%) relied on self-report and survey data. Following this, 28.35% used observational data (including reaction times), 11.03% used biological or physiological data, and the remaining 5.30% used other types of data or methodologies (for example, they used computational modelling or presented a new method) to answer their primary research question.

However, it does not appear that studies using different types of data are any more or less prone to invoking theory. Of the studies that used self-report and survey data, 26.16% named a specific theory. Of the studies that used biological and physiological data, 19.77% named a specific theory. Of the studies that used observational or behavioural data, 22.20% named a specific theory. Of the studies that used other types of data, 25.88% named a specific theory. See S5 and S6 Tables in S1 File for complete counts.

Further, it does not appear that theoretically derived predictions are more conducive to any specific type of study. Only 17.36% of studies using self-report data, 11.86% of studies using biological/physiological data, 11.87% of studies using observational data, and 20% of studies using other types of data explicitly claimed to be testing theoretically derived predictions.

## Question 5: How many theories were mentioned in this 11-year period?

We also counted the number of theories that were mentioned or referred to explicitly in each of the 2,225 manuscripts. As described in the S1 File, slightly different criteria were used for this task so as to be as inclusive as possible. A total of 359 theories were mentioned in text over the 11-year period. Most theories were mentioned in only a single paper (*mode* = 1, *median* = 1, *mean* = 1.99). The full list of theories is presented in S7 Table in S1 File. For ease of reference, the top 10 most-mentioned theories are displayed below in Table 1.

## Exploratory analysis: Did authors use the word *hypothesis* in place of *theory*?

One concern may be that authors are misusing the word *hypothesis* to refer to these formal, higher-level *theories*. That is, that authors are using the word *hypothesis* when they should be

**Table 1. The top 10 most mentioned theories sorted according to the total number of mentions.**

| Name | 2009 | 2010 | 2011 | 2012 | 2013 | 2014 | 2015 | 2016 | 2017 | 2018 | 2019 | Total |
|---|---|---|---|---|---|---|---|---|---|---|---|---|
| Signal Detection Theory | 2 | 2 | 2 | 3 | 2 | 6 | 1 | 2 | 3 | 2 | 2 | 27 |
| Prospect Theory (Also Cumulative Prospect) | 3 | | 2 | 2 | | 4 | 2 | 2 | 2 | 1 | 3 | 21 |
| Attachment Theory | 1 | 5 | | 2 | 2 | | | 3 | 2 | 2 | | 17 |
| Life History Theory | | 2 | | | 3 | 2 | 1 | 4 | | 2 | 1 | 15 |
| Construal-Level Theory (Psychological Distance) | 2 | | | 4 | 2 | 2 | 3 | | 1 | | | 14 |
| Social-Identity Theory | 2 | 2 | 2 | 2 | 1 | 1 | | | | 3 | | 13 |
| System Justification Theory | 1 | 2 | 1 | 1 | 3 | 1 | 2 | | | | 1 | 12 |
| Game Theory | | | | 1 | 1 | 5 | 1 | 2 | 1 | | | 11 |
| Item Response Theory | | | 2 | 1 | | 4 | 1 | | 1 | 1 | 1 | 11 |
| Self-Affirmation Theory | 2 | | 2 | 2 | 1 | 1 | | 1 | | 1 | | 10 |
| Terror Management Theory | 1 | | 2 | 1 | 1 | 3 | 1 | 1 | | | | 10 |

using the word *theory*. To examine this possibility, we mined all 2,225 documents for the word *hypothesis* and examined the immediate context surrounding each instance.

If the authors were referring to a formally named, superordinate hypothesis derived from elsewhere (e.g., if it satisfied the criteria for a theory) it was coded as 1. It was coded as 0 if the authors were using *hypothesis* correctly. Specifically, it received a code of 0 if the authors were referring to their own hypothesis or expectations (e.g., *our* hypothesis, *this* hypothesis, etc), if they were describing a statistical analysis (e.g. *null* hypothesis), or if they were describing an effect or pattern of results (e.g., the hypothesis *that*. . .). Instances in the references were not counted. Two independent coders rated each instance of the word. Initial coder agreement was 89.5% and significantly greater than chance (Fleiss' kappa = .61, $p < .001$). As before, after initial coder agreement was analysed, a third coder resolved any disagreements and the final ratings (consisting of scores for which at least two coders agreed) were analysed.

Of the 2225 articles published over the 11 years, 62% used the word hypothesis (n = 1,386). Of those, 14.5% (n = 202) used hypothesis in a way to refer to a larger, formal, or externally derived *theory*. Put differently, this constitutes 9% of the total corpus ($N$ = 2,225). Complete counts according to year are displayed in S8 Table in S1 File. Thus, it appears that this misuse of the word is not very common. However, even if we were to add this total count to our previous analysis of *theory*, it would not change our overall interpretation: the majority of papers published in *Psychological Science* are not discussing nor relying on theories in their research.

## Discussion

The *Psychological Science* website states that "The main criteria for publication in *Psychological Science* are general theoretical and empirical significance and methodological/statistical rigor" [30, 31]. Yet, only 53.66% of articles published used the word *theory*, and even fewer named or claimed to test a specific theory. How can research have general theoretical significance if the word theory is not even present in the article?

A more pressing question, perhaps, is how can a field be contributing towards cumulative theoretical knowledge if the research is so fractionated? We identified 359 psychological theories that were referred to in-text (see S7 Table in S1 File for the complete list) and most of these were referred to only a single time. A recent review referred to this as *theorrhea* (a mania for new theory), and described it as a symptom stifling "the production of new research" [38]. Indeed, it's hard to imagine that a cumulative science is one where each theory is examined so infrequently. One cannot help but wonder how the field can ever move towards theoretical consensus if everyone is studying something different—or, worse, studying the same thing with a different name.

These data provide insight into how psychologists are using psychological theories in their research. Many papers made no reference to a theory at all and most did not explicitly derive their predictions from a theory. It's impossible to know why a given manuscript was written in a certain way, but we offer some possibilities to help understand why some authors neglected to even include the word *theory* in their report. One possibility is that the research is truly a-theoretical or descriptive. There is clear value in descriptive research—value that can ultimately amount to theoretical advancement [17, 18, 20] and it would be misguided to avoid interesting questions because they did not originate from theory.

It's also possible that researchers are testing auxiliary assumptions [39] or their own interpretations (instead of the literal interpretations or predictions) of theories [40]. This strategy is quite common: authors describe certain effects or qualities of previous literature (e.g., the literature review) in their introduction to narrate how they developed a certain hypothesis or idea, then they state *their own* hypothesis. Such a strategy is fine, but certainly does not amount to a

quantifiable prediction derived from a pre-specified theory. Further, given that psychological theories are almost always verbal [2, 15], there may not even be literal interpretations or predictions to test.

An additional possibility is that researchers may be focusing on "effects" and paradigms rather than theories per se. Psychology is organized topically—development, cognition, social behaviour, personality—and these topics are essentially collections of effects (e.g., motivated reasoning, the Stroop effect, etc). Accordingly, researchers tend to study specific effects and examine whether they hold under different conditions. Additionally, a given study may be conducted because it is the logical follow-up from a previous study they conducted, not because the researchers are interested in examining whether a theory is true or not.

However, it's also important to consider the qualities of the research that *did* use the word theory and why. Recall only 33.58% of articles using the word *theory* or *theories* said anything substantial about a theory. For the remaining articles, it's possible that these words and phrases were injected post-hoc to make the paper seem theoretically significant, because it is standard practice, or because it is a journal requirement. That is, this may be indicative of a specific type of HARKing: searching the literature for relevant hypotheses or predictions after data analysis, known as RHARKing [29]. For example, some researchers may have conducted a study for other reasons (e.g., personal interest), but then searched for a relevant theory to connect the results to after the fact. It's important to note that HARKing can be prevented by preregistration, but preregistration was only used in 11.11% of the papers that claimed to test a theory. Of course, it's impossible to know an author's motivation in absence of a preregistration, but the possibility remains quite likely given that between 27% and 58% of scientists admit to HARKing [29].

Finally, this data provides insight into the kind of research psychologists are conducting. The majority (55.26%) is conducted using self-report and survey data. Much less research is conducted using observational (28.35%) and biological or physiological (11.03%) data. While not as bleak as a previous report claiming that behavioural data is *completely* absent in the psychological sciences [41], this points to a limitation in the kinds of questions that can be answered. Of course, self-report data may be perfectly reasonable for some questions, but such questions are necessarily restricted to a narrower slice of human behaviour and cognition. Further, a high degree of reliance on a single method certainly contrasts with the large number of theories being referenced. It is worth considering how much explanatory power each of the theories have if most of them are discussed exclusively in the context of self-report and survey data.

## Limitations and additional considerations

The present results describe only one journal: *Psychological Science.* However, we chose this journal because it is one of the top journals in the field, because it publishes research from all areas of psychology, and because it has explicit criteria for theoretical relevance. Thus, we expected that research published in this journal would be representative of some of the theoretically relevant research being conducted. So, we do not claim that the results described here statistically generalize to other journals, only that they describe the pattern of research in one of the top journals in psychology. One specific concern is that *Psychological Science* limits articles to 2,000 words, and this may have restricted the ability to describe and reference theories. This may be true, though would seem that the body of knowledge a piece of research is contributing towards would be one of the most important pieces of information to include in a report. That is, if the goal of that research were to contribute to cumulative knowledge, it does not require many words to refer to a body of theory by name.

An additional concern may be that, in some areas of psychology, "theories" may be referred to with a different name (e.g., *model* or *hypothesis*). However, the terms *model* and *hypothesis* do not carry the formal weight that *scientific theory* does. In the hierarchy of science, *theories* are regarded as being the highest status a claim can achieve—that most articles use it casually and conflate it with other meanings is problematic for clear scientific communication. In contrast, *model* or *hypothesis* could be used to refer to several different things: if something is called *model*, then it's not claiming to be a *theory*. Our additional analysis only identified a small minority of papers that used *hypothesis* in this fashion (9% of the total corpus). While this number is relatively small, this does highlight an additional issue: the lack of consistency with which *theories* are referred to and discussed. It is difficult and confusing to consistently add to a body of knowledge if different names and terms are used.

Another claim might be that theory should simply be implicit in any text; that it should permeate through one's writing without many direct references to it. If we were to proceed in this fashion, how could one possibly contribute to cumulative theory? If theory need not be named, identified, or referred to specifically, how is a researcher to judge what body of research they are contributing to? How are they to interpret their findings? How is one even able to design an experiment to answer their research question without a theory? The argument has been made that researchers need theory to guide methods [5, 6, 9]—this is not possible without, at least, clearly naming and referencing theories.

A final limitation to note is one regarding the consistency of the coders. While the fair to moderate kappas obtained here may seem concerning at first, we believe this reflects the looseness and vaguery with which words like *theory* are used. Authors are often ambiguous and pad their introductions and discussion with references to models and other research; it is not often explicit whether a model is simply being mentioned or whether it is actually guiding the research. Further complicating things is that references to theories are often inconsistent. Thus, it can be a particularly difficult task to determine whether an author actually derived their predictions from a specific theory or whether they are simply discussing it because they later noted the similarities. Such difficulties could have contributed to the lower initial agreement among coders. Therefore, along with noting that the kappas are lower than would be ideal, we also suggest that future researchers should be conscious of their writing: it's very easy to be extremely explicit about where one's predictions were derived from and why a test is being conducted. We believe this to be a necessary component of any research report.

## Concluding remarks

Our interpretation of this data is that the published research we reviewed is simultaneously saturated and fractionated, and theory is not guiding the majority of research published in *Psychological Science* despite this being the main criteria for acceptance. While many articles included the words *theory* and *theories*, these words are most often used casually and non-specifically. In a large subset of the remaining cases, the theoretical backbone is no more than a thin veneer of relevant rhetoric and citations.

These results highlight many questions for the field moving forward. For example, it's often noted that psychology has made little progress towards developing clearly specified, cumulative theories [2, 15] but what should that progress look like? What is the role of theory in psychological science? Additionally, while it is widely assumed that psychological research follows the hypothetico-deductive model, these data suggest this is not necessarily the case. There are many other ways to do research and not all of them involve theory testing. If the majority of research in a top journal is not explicitly testing predictions derived from theory, then perhaps it exists to explore and describe interesting effects. There is certainly nothing wrong with a

descriptive approach, and this aim of psychology has been suggested for at least half a century [20, 42, 43].

To be clear, we are not suggesting that every article should include the word *theory*, nor that it should be a requirement for review. We are not even suggesting that research *needs to be* based in theory. Instead, we are simply pointing out the pattern of research that exists in one of the leading research journals with the hope that this inspires critical discussion around the process, aims, and motivation of psychological research. There are many ways to do research. If scientists want to work towards developing nomothetic explanations of human nature then, yes, theory can help. If scientists simply want to describe or explore something interesting, that's fine too.

## Supporting information

**S1 File.**
(DOCX)

## Author Contributions

**Conceptualization:** Jonathon McPhetres.

**Data curation:** Jonathon McPhetres.

**Formal analysis:** Jonathon McPhetres.

**Investigation:** Jonathon McPhetres, Nihan Albayrak-Aydemir, Ana Barbosa Mendes, Elvina C. Chow, Patricio Gonzalez-Marquez, Erin Loukras, Annika Maus, Aoife O'Mahony, Christina Pomareda, Maximilian A. Primbs, Shalaine L. Sackman, Conor J. R. Smithson, Kirill Volodko.

**Methodology:** Jonathon McPhetres.

**Project administration:** Jonathon McPhetres.

**Supervision:** Jonathon McPhetres.

**Visualization:** Jonathon McPhetres.

**Writing – original draft:** Jonathon McPhetres.

**Writing – review & editing:** Jonathon McPhetres, Nihan Albayrak-Aydemir, Ana Barbosa Mendes, Elvina C. Chow, Patricio Gonzalez-Marquez, Erin Loukras, Annika Maus, Aoife O'Mahony, Christina Pomareda, Maximilian A. Primbs, Shalaine L. Sackman, Conor J. R. Smithson, Kirill Volodko.

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
