## [Decision Letter · Decision Letter 0]

21 Dec 2020

PONE-D-20-28543

A decade of theory as reflected in Psychological Science (2009-2019)

PLOS ONE

Dear Dr. McPhetres,

Thank you for submitting your manuscript to PLOS ONE. After careful consideration, we feel that it has merit but does not fully meet PLOS ONE’s publication criteria as it currently stands. Therefore, we invite you to submit a revised version of the manuscript that addresses the points raised during the review process.

I would like to thank you for this submission, I speak for myself and hopefully both reviewers when I say it was a pleasure to read and looks to be a strong and impactful addition to the literature. I agree with the reviewers though that before acceptance some minor additions are required. There was no major conflicts between what the reviewers said, in fact I personally think they compliment each other quite well, so that addressing their concerns in one area should help the entire work overall. While I think that all the suggestions are valid, and they are not too onerous on you and your teams, I believe the most critical issues addressing reviewer 1's statements about the usage of the term "hypothesis", as they correctly point out how your lack of capture in your coding regime may overestimate the atheoretical nature of the field. I would also stress the need to address reviewers 2's concerns about the agreement scores between coders. Once these issues are addressed I believe this work will be strong enough to publish. I understand that, with the upcoming holidays for many institutions as well as Covid restrictions, it may be difficult for you and your team to address all these concerns quickly, therefore while I suggested approximately 48 days as time for resubmission (slightly more than the 45 that is typical), if you require more time please contact us and we can extend this deadline. We all understand that the current pace of the academic and non-academic world is not typical, and we do not want you or your time to feel constrained by this timeline. I thank you for your submission

We look forward to receiving your revised manuscript.

Kind regards,

T. Alexander Dececchi, Ph.D

Academic Editor

PLOS ONE

Journal Requirements:

2. We note the study analyses publications from a single publication (Psychological Science) as part of this study. We note that you have acknowledged this as a limitation in the Discussion, and indicate that "we do not claim that the results described here statistically generalize to other journals".

However, some of the conclusions made do appear to suggest the results are generalizable to wider group, e.g. "We interpret this to suggest that most psychological research is not driven by theory, nor can it be contributing to cumulative theory building."

Please revise accordingly. This is required in order to meet PLOS ONE's 4th publication criterion, which states that 'Conclusions are presented in an appropriate fashion and are supported by the data.'

https://journals.plos.org/plosone/s/criteria-for-publication#loc-4

Additional Editor Comments:

First off I would like to apologize for the delays and I thank you for your understanding and patiences. Second I wish to congratulate you on an overall very compelling and informative study. This line of inquiry is needed to help drive psychological research forward. That said I also agree with the reviewers on their most significant suggestions, especially the omission of "hypothesis" from your analysis as brought up by reviewer 1 and the moderate coder agreement scores brought forward by reviewer 2. I believe addressing these in the next version will greatly improve it and make it even more accessible to wider audience. I thank you all for this manuscript and I look forward to your re-submission.

Reviewers' comments:

Reviewer's Responses to Questions

**Comments to the Author**

1. Is the manuscript technically sound, and do the data support the conclusions?

Reviewer #1: Yes

Reviewer #2: Partly

2. Has the statistical analysis been performed appropriately and rigorously? 

Reviewer #1: Yes

Reviewer #2: Yes

3. Have the authors made all data underlying the findings in their manuscript fully available?

Reviewer #1: Yes

Reviewer #2: Yes

4. Is the manuscript presented in an intelligible fashion and written in standard English?

Reviewer #1: Yes

Reviewer #2: Yes

5. Review Comments to the Author

Reviewer #1: Thank you for the opportunity to review this manuscript. The authors have chosen a fascinating topic and approached it in an elegant and innovative fashion. Their analytic approach is well considered and, given the constraints of their analysis, the conclusions they draw from their results are sound. I certainly believe that this submission contributes in a unique and meaningful way to the literature on theorizing in psychology, and I think it would make a fine addition to your outlet.

My only concern regards the authors’ decision to omit the term “hypothesis” from their analyses. Although the authors go to some lengths to justify this decision, I remain unconvinced by their argument. Anecdotally, I think it is common in the field to refer to theory and hypothesis interchangeably, and there are certain types of hypothesis that satisfy the sort of superordinate status the authors ascribe to “theory”. In the evolutionary psychological literature, for example, theories such as inclusive fitness are referred to as “first order hypotheses”, from which subsidiary, testable hypotheses or predictions can be derived. In a similar vein, it is widely recognised that psychology progresses via cumulative tests of lower-order hypotheses derived from higher-order theories – here, it is quite reasonable to expect that researchers only explicitly refer to the former (i.e., the subject of their analysis), rather than the broader theoretical framework from which their hypotheses are derived; nevertheless, progressive empirical support for lower-order hypotheses constitutes cumulative support for higher-order theories. In short, these two terms cannot be readily individuated. On the other hand, I am sympathetic to the fact that a hypothesis can also refer to its more trivial sense (i.e., specific, testable predictions), which would require a more nuanced, qualitative analysis and coding of target articles to differentiate the more substantive use of the term (i.e., theory) from its more trivial form (i.e., empirical predictions). Nevertheless, I believe that such an analysis is required to demonstrate, convincingly, whether psychological science operates in the atheoretical manner the authors describe.

Otherwise, another, minor suggestion is that the authors might like to consider complementing some of their results with inferential analyses (e.g., chi-square analyses), where appropriate. It would be interesting to see whether the differences they cite reach statistical significance.

In closing, I would like to congratulate the authors on a fascinating submission, and I wish them all the best in their future endeavours.

Reviewer #2: Overview: This manuscript explored mentions of theory in the past 10 years in the journal Psychological Science. This paper attempts to provide an answer about the extent to which modern psychological research is guided by theory. This manuscript is innovative, clever, and overall well-written. The authors present interesting findings about psychological research’s current lack of grounding in theory without necessarily prescribing a need for change. My primary concern is the low agreement between coders on what constitutes a reference to theory, as captured by the Fleiss’ kappas. While these values suggest coders agreed at better than chance rates, their agreement was only fair to moderate at best. This goes back to the authors’ question of how to identify a theory and thus a reference to theory. More detail and explanation for these low agreement scores is needed.

This manuscript examined

1. It would be helpful to readers to include the theories that were mentioned most often in the text section on how many theories were mentioned in addition to the supplemental information.

2. Psych Science article’s introduction and discussion sections are limited to 2000 words. The authors might consider whether this word limit could have contributed to lower rates of including references to theory.

3. The manuscript currently lacks information to interpret Fleiss’ kappa according to cut points (i.e., no agreement, slight agreement, fair agreement, etc.) to help the reader better understand the level of agreement between coders. Furthermore, according to cut points for Fleiss’ kappa, coders showed only moderate agreement for the initial question of referring to a specific theory and only fair agreement for testing a prediction from a specific theory. These low kappas are concerning. The authors should note this is a limitation and offer potential explanations for why coders showed these levels of disagreement. It would help to contextualize the kappas based on what other studies using this as a measure of agreement have found.

6. PLOS authors have the option to publish the peer review history of their article (what does this mean?). If published, this will include your full peer review and any attached files.

Reviewer #1: No

Reviewer #2: No

---

## [Author Response · Author response to Decision Letter 0]

11 Jan 2021

Response to comments

Editor comments

Response: I have reviewed the guidelines and believe that my files now satisfy these requirements. 

2. We note the study analyses publications from a single publication (Psychological Science) as part of this study. We note that you have acknowledged this as a limitation in the Discussion, and indicate that "we do not claim that the results described here statistically generalize to other journals".

However, some of the conclusions made do appear to suggest the results are generalizable to wider group, e.g. "We interpret this to suggest that most psychological research is not driven by theory, nor can it be contributing to cumulative theory building."

Please revise accordingly. This is required in order to meet PLOS ONE's 4th publication criterion, which states that 'Conclusions are presented in an appropriate fashion and are supported by the data.'

Response: I have removed, to the best of my knowledge, statements that imply a generalisation to all of psychology. For example, I have reworded the statement you pointed out to read “that the research published in this flagship journal is not driven by theory”, in the beginning of the concluding remarks “the published research we reviewed” and “theory is not guiding the majority of research published in Psychological Science.” 

Reviewer Comments

Reviewer #1: Thank you for the opportunity to review this manuscript. The authors have chosen a fascinating topic and approached it in an elegant and innovative fashion. Their analytic approach is well considered and, given the constraints of their analysis, the conclusions they draw from their results are sound. I certainly believe that this submission contributes in a unique and meaningful way to the literature on theorizing in psychology, and I think it would make a fine addition to your outlet.

Response: Thank you for the positive evaluation of our work.

1. My only concern regards the authors’ decision to omit the term “hypothesis” from their analyses. Although the authors go to some lengths to justify this decision, I remain unconvinced by their argument. Anecdotally, I think it is common in the field to refer to theory and hypothesis interchangeably, and there are certain types of hypothesis that satisfy the sort of superordinate status the authors ascribe to “theory”. In the evolutionary psychological literature, for example, theories such as inclusive fitness are referred to as “first order hypotheses”, from which subsidiary, testable hypotheses or predictions can be derived. In a similar vein, it is widely recognised that psychology progresses via cumulative tests of lower-order hypotheses derived from higher-order theories – here, it is quite reasonable to expect that researchers only explicitly refer to the former (i.e., the subject of their analysis), rather than the broader theoretical framework from which their hypotheses are derived; nevertheless, progressive empirical support for lower-order hypotheses constitutes cumulative support for higher-order theories. In short, these two terms cannot be readily individuated. On the other hand, I am sympathetic to the fact that a hypothesis can also refer to its more trivial sense (i.e., specific, testable predictions), which would require a more nuanced, qualitative analysis and coding of target articles to differentiate the more substantive use of the term (i.e., theory) from its more trivial form (i.e., empirical predictions). Nevertheless, I believe that such an analysis is required to demonstrate, convincingly, whether psychological science operates in the atheoretical manner the authors describe.

Response: We have included this additional analysis. I now detail the results in the “exploratory analysis” section and have included a table detailing this data by year. The results show that, while some people do use hypothesis in place of theory, this is a minority of papers (only 9% of the total corpus). 

2. Otherwise, another, minor suggestion is that the authors might like to consider complementing some of their results with inferential analyses (e.g., chi-square analyses), where appropriate. It would be interesting to see whether the differences they cite reach statistical significance.

Response: We have not included inferential statistics because we have analysed the entire corpus of articles. Thus, there is no ‘population’ to generalise our results to with the interpretation of a p-value That is, because we have all the articles, everything is an actual difference if the numbers differ in their absolute value when you have the whole population and no p-values are needed to determine whether the numbers would differ significantly given a frequentist methodology and interpretation (e.g. what would happen if we repeated the study 100 times). 

3. In closing, I would like to congratulate the authors on a fascinating submission, and I wish them all the best in their future endeavours.

Response: Thank you again for your constructive feedback!

Reviewer #2: Overview: This manuscript explored mentions of theory in the past 10 years in the journal Psychological Science. This paper attempts to provide an answer about the extent to which modern psychological research is guided by theory. This manuscript is innovative, clever, and overall well-written. The authors present interesting findings about psychological research’s current lack of grounding in theory without necessarily prescribing a need for change. My primary concern is the low agreement between coders on what constitutes a reference to theory, as captured by the Fleiss’ kappas. While these values suggest coders agreed at better than chance rates, their agreement was only fair to moderate at best. This goes back to the authors’ question of how to identify a theory and thus a reference to theory. More detail and explanation for these low agreement scores is needed.

Response: Thanks for pointing this out- I think this is the result of a miscommunication on my part. I have included additional text in the methods section to clarify how the data were coded by raters and why I do not believe the fair-to-moderate kappas to be a problem. I will also explain a bit more here, though.

 First, just to clarify, the coding took place in two stages. Initially, two coders independently reviewed each article and recorded ratings. The kappa reported in the article was computed on this initial coding only. Then, in the second step, a third coder reviewed the disagreements and it was the ratings after this final round of coding which we analyse. So, the resulting code that we analysed for the main results was the result of codes on which one of the two conditions were satisfied: either a) two coders agreed 100% or b) two out of three coders agreed 100%.

 This means that that lower level of agreement was corrected when the third coder independently reviewed the disagreements (ie the kappa doesn’t necessarily describe the data we analysed).

 Thus, I do not think this is an issue because 1) agreement wasn’t too bad to begin with, it was still at moderate levels for the more complicated ratings, 2) more categories means lower agreement, and 3) the tie-breaker means the ratings are the result of agreement by at least two coders.

In response, I have made the following changes to the manuscript.

On page 3-4 where I describe the coding, I have reworded this to read as follows. Second, I have included some brief rules of thumb on pages 3-4. It now reads as follows:

“Each article was initially scored independently by two individual coders who were blind to the purpose of the study; Fleiss’ Kappa is report for this initial coding. Recommendations suggest that a kappa between .21-.40 indicates fair agreement, .41-.60 indicates moderate agreement, .61-.80 indicates substantial agreement, and .81-1.0 is almost perfect agreement (37). 

After the initial round of coding, two additional blind coders and the first author each independently reviewed a unique subset of disagreements to resolve ties. This means that the ratings we analyse in the following section are the result of codes only for which two independent raters (or two out of three raters) agreed 100%. ”

This manuscript examined

1. It would be helpful to readers to include the theories that were mentioned most often in the text section on how many theories were mentioned in addition to the supplemental information.

Response: I have noted this in under the heading of “ Question 5: How many theories were mentioned…” (page 8). I have included a table with the top-10 most mentioned theories. 

2. Psych Science article’s introduction and discussion sections are limited to 2000 words. The authors might consider whether this word limit could have contributed to lower rates of including references to theory.

Response: Good point. At the beginning of the limitations sections I have stated the following:

“One specific concern is that Psychological Science limits articles to 2,000 words, and this may have restricted the ability to describe and reference theories. This may be true, though would seem that the body of knowledge a piece of research is contributing towards would be one of the most important pieces of information to include in a report. That is, if the goal of that research were to contribute to cumulative knowledge it, it does not require many words to refer to a body of theory by name.”

3. The manuscript currently lacks information to interpret Fleiss’ kappa according to cut points (i.e., no agreement, slight agreement, fair agreement, etc.) to help the reader better understand the level of agreement between coders. Furthermore, according to cut points for Fleiss’ kappa, coders showed only moderate agreement for the initial question of referring to a specific theory and only fair agreement for testing a prediction from a specific theory. These low kappas are concerning. The authors should note this is a limitation and offer potential explanations for why coders showed these levels of disagreement. It would help to contextualize the kappas based on what other studies using this as a measure of agreement have found.

Response: I have included the rules of thumb for kappa in the methods section, as described earlier. This is related to my previous response regarding the calculation of the kappas. Namely, we only used the codes for which two raters agreed (e.g. after tie-breaking). However, there are a few other practical considerations to be made here. 

 First, these categories are extremely difficult to code. They may seem straightforward but 1) authors are often extremely vague, 2) we are coding something for which we expect there to be misuses of the word (which adds noise), and 3) coding multiple categories will necessarily reduce agreement. 

 The coders did almost perfectly when coding whether a study was pre-registered- had this not been the case, I would have been more concerned about the other categories. 

Going into this project, I initially thought it would be straightforward to identify what a theory is, but it is not. People use this word so loosely that it makes any coding scheme feel inadequate. Authors contradict themselves and make ambiguous statements. I think this has more to do with the articles rather than the coders or the coding scheme. Some of these thoughts were already in the manuscript though. And I’m hesitant to put all of these thoughts into the paper, but I have added some discussion on this to the end of the limitations section.

---

## [Decision Letter · Decision Letter 1]

18 Feb 2021

A decade of theory as reflected in  Psychological Science (2009-2019)

PONE-D-20-28543R1

Dear Dr. McPhetres

We’re pleased to inform you that your manuscript has been judged scientifically suitable for publication and will be formally accepted for publication once it meets all outstanding technical requirements.

Kind regards,

T. Alexander Dececchi, Ph.D

Academic Editor

PLOS ONE

Additional Editor Comments (optional):

After reading your revisions the reviewers and myself all agree that we should accept your manuscript. Congratulations. I know this was a long time in the works and I apologize for that. I thank you for your patience

Reviewers' comments:

Reviewer's Responses to Questions

**Comments to the Author**

1. If the authors have adequately addressed your comments raised in a previous round of review and you feel that this manuscript is now acceptable for publication, you may indicate that here to bypass the “Comments to the Author” section, enter your conflict of interest statement in the “Confidential to Editor” section, and submit your "Accept" recommendation.

Reviewer #1: All comments have been addressed

Reviewer #2: All comments have been addressed

2. Is the manuscript technically sound, and do the data support the conclusions?

Reviewer #1: Yes

Reviewer #2: Partly

3. Has the statistical analysis been performed appropriately and rigorously? 

Reviewer #1: Yes

Reviewer #2: Yes

4. Have the authors made all data underlying the findings in their manuscript fully available?

Reviewer #1: Yes

Reviewer #2: Yes

5. Is the manuscript presented in an intelligible fashion and written in standard English?

Reviewer #1: Yes

Reviewer #2: Yes

6. Review Comments to the Author

Reviewer #1: The authors have done a fine job responding to the reviewers' concerns. I wish them all the best in their future endeavours.

Reviewer #2: The authors did a great job incorporating reviewer feedback into the revised document. The only other change I would suggest is tempering some of the strong language in the abstract and discussion somewhat to be more suggestive of potential implications of the findings. For example, in the abstract it states: “We interpret this to suggest that the majority of research published in this flagship journal is not driven by theory, nor can it be contributing to cumulative theory building.” Maybe instead say something like, “Given that the majority of research published in this flagship journal does not derive specific hypotheses from theory, we suggest that theory is not a primary driver of much of this research. Further, the research findings themselves may not be contributing to cumulative theory building“. From what I understand of the findings, several studies did reference theory, they just did not use theory to specifically derive their hypotheses.

7. PLOS authors have the option to publish the peer review history of their article (what does this mean?). If published, this will include your full peer review and any attached files.

Reviewer #1: No

Reviewer #2: No

---

## [Editor Report · Acceptance letter]

25 Feb 2021

PONE-D-20-28543R1 

A decade of theory as reflected in
*Psychological Science* (2009-2019) 

Dear Dr. McPhetres:

I'm pleased to inform you that your manuscript has been deemed suitable for publication in PLOS ONE. Congratulations! Your manuscript is now with our production department. 

Kind regards, 

on behalf of

Dr. T. Alexander Dececchi 

Academic Editor

PLOS ONE